# Assessing the effectiveness of fuzzy logic-based models for predicting sports event outcomes: A CRITIC-VIKOR approach

**Taibo Liu** [ORCID] *

Department of Basic Education, Jiangsu Shipping College, Nantong, China

* liutb2024@163.com

## Abstract

Incorporating fuzzy logic-based models into sports prediction has generated significant interest due to the intricate nature of athletic events and the many factors influencing their outcomes. This study evaluates the effectiveness of fuzzy logic-based models in predicting sports event outcomes using a hybrid CRITIC-VIKOR approach. The objective is to improve the accuracy and reliability of sports predictions by addressing the complexity and uncertainty inherent in sports data. The study utilizes a comprehensive dataset comprising historical data on team performance, player statistics, and other relevant factors influencing sports outcomes. The CRITIC method determines each criterion's importance, while the VIKOR method ranks the predictive models to identify the optimal choice. Key findings indicate that the proposed hybrid approach significantly enhances the precision of predictions compared to traditional methods. The best-performing model identified through this approach provides reliable decision support for sports analysts, coaches, and managers. The study recommends incorporating this integrated model into sports analytics for better team management and sports betting decision-making.

**Data Availability Statement:** All relevant data are within the paper.

**Funding:** The author(s) received no specific funding for this work.

## 1. Introduction

Sports forecasting has long been a fascinating topic of study for sportspeople, managerial staff, players, and an increasing number of bettors. Fuzzy logic is currently receiving increased attention from organizations looking to predict athletic events. The study of using fuzzy logic-based models to predict the outcomes of athletic events combines logical reasoning, data analysis, and sports analytics. The way to deal with ambiguity and unclear data, the fuzzy logic approach to mathematics assigns several categories with varying degrees of involvement. Since inconsistencies and fuzziness are so prevalent in sports-related data, this makes it feasible to acquire and understand that data. Due to the complexity of sporting events and the numerous variables that might affect their results, the application of fuzzy logic-based models in sports prediction has attracted a lot of interest. The creation and implementation of predictive models have also been influenced by sports analytics, which is the utilization of data and statistical techniques to acquire an understanding and make sensible judgments in the field of sports.

**Competing interests:** The authors have declared that no competing interests exist.

Fuzzy logic-based models are used to forecast the results of sporting events for a variety of reasons. Increasing the precision and dependability of result forecasts is one of the main goals. Fuzzy logic-based models may produce forecasts that take into account uncertainty and take into account the complexity of sporting events by taking into account a wide variety of pertinent factors to decision-making. These models can also assist decision-makers who are involved in the sports sector. Trainers, club executives, sports specialists, and even sports bettors can benefit from the information provided by fuzzy logic-based models. By identifying the most crucial factors affecting sports results and providing the best feasible tactics or game strategies, the models may assist with choices about selecting a team, the player moves, and game planning. In general, the use of fuzzy logic-based models to forecast the results of sporting events constitutes a multifaceted topic that integrates sports analytics, data analysis techniques, and fuzzy logic concepts. It strives to increase precision in forecasting, offer decision support, and deepen knowledge of the variables affecting sports results in the ever-evolving and cutthroat world of sports. The purpose of the study is to investigate how fuzzy logic-based models may be used to forecast sporting event results. It also looks into how well the CRITIC and VIKOR approaches work for rating and assessing the efficacy of the prediction models. The endeavor aims to give insights into the decision-making process and improve the precision and dependability of sports event forecasts by integrating various methodologies.

Support vector machines (SVMs) are useful instruments for dealing with issues involving classification. The SVM algorithms' drawback is their inability to create rules. The hybrid fuzzy-SVM model (HFSVM) in the proposed study [1], which combines the fuzzy approach and SVM methodology, may be used to predict basketball game results and help instructors, squads, and athletes improve their game. The HFSVM paradigm has the capability to generate rules using fuzzy membership processes, which serves as a distinctive feature of the fuzzy technique since it integrates the beneficial effects of the fuzzy strategy with the SVM methodology. The experimental findings show that when the HFSVM paradigm and the SVM model are evaluated, the HFSVM paradigm not only performs superior to the SVM paradigm but also offers predictability that is generally recognized as acceptable. Given this, adopting the HFSVM paradigm to examine basketball game data may produce useful discoveries. Sports statistics has grown in popularity over the past several years as a tool to assess player and club performance, forecast game outcomes across a variety of sports, and aid hiring choices. With the use of sports statistics, football, one of the oldest sports, is updating its methods. Saricaoğlu et al. [2] goal is to use automated machine learning approaches to develop a football game forecasting system for the Turkish Super League (TSL). Game outcome forecasting methods were created utilizing statistical methods such as logistic analysis, linear and quadratic differential assessments, K-nearest neighbors, support vector machines, and random forests on the basis of the TSL information collected over the previous few years. It is suggested that one use a set of ten models made with seven distinct building methods.

Among the tournaments with the most coverage in foreign media is the English Premier League (EPL). The suggested study gives an innovative technique for forecasting EPL game results. Based on how crucial they were to the game, the 15 characteristics were chosen. To anticipate the results of contests, six distinct SVM-based paradigm differences—linear, quadratic, cubic, fine radial basis function, medium radial basis function, and coarse radial basis function—were created. The effectiveness of the top designs constructed from a second batch of brand-new data was evaluated using a five-step cross-validation method. The proposed study shows that the linear SVM framework can predict both cultivated and untreated data with a high forecasting accuracy of 100%. One can infer that the strategy adopted and the qualities picked could lead to a precise estimation of the results of games involving the best six EPL sides [3]. To estimate football game results, Rahman [4] suggested an effective architecture

based on deep neural networks (DNNs) and artificial neural networks (ANNs). It uses a dataset with extra data like teamwork metrics and historical international football game outcomes. The athletic data is evaluated and processed using ANN and DNN to get a forecast score. Different datasets serve for training, evaluation, and verification. According to the suggested DNN design, the appropriate algorithm was able to predict the Football Cup 2018 games incredibly accurately. The framework correctly forecast 63.3% of the games. With the necessary datasets and more precise team data, the precision may be improved. According to this, athletic events like football games might potentially be accurately predicted using deep learning. If history information on each squad, player, and game were made available, the predictor would perform better and be more realistic.

It is now more vital than ever to employ technology to predict the results of sporting events since both the market for gambling and technology have advanced tremendously. Analyzing large amounts of data has a number of restrictions. However, by employing methods of AI, this problem may be solved. Given the volume of data that must be considered, sports provide an excellent illustration of an AI's difficulty. The suggested paper offers a summary of studies leveraging different types of AI to forecast sports matches. several techniques for forecasting results were discovered via the examination of several sports, including basketball, javelin, football, soccer, and horse racing. Ultimately, a structure for developing an AI-based system to forecast football game results is offered, taking into consideration the recently published assessment [5]. In this work, various fuzzy logic-based models for forecasting sports results are evaluated and compared. This study compares multiple models in an effort to find the best precise and efficient method for forecasting the outcomes of sporting events by weighing the derived criteria.

The primary objectives of this research are as follows:

- To analyze the accuracy and reliability of fuzzy logic-based models in forecasting sports event results and assess their suitability for sports predictions.

- To identify the pertinent decision-making factors that exert significant influence on sports event outcomes.

- To utilize the CRITIC technique for the weighting and prioritization of relevant and comparable factors in the prediction of sports outcomes.

- To apply the VIKOR method to evaluate and select the most suitable fuzzy logic-based models for predicting sports event results.

- To elucidate the potential applications of fuzzy logic-based models in predicting sports event outcomes.

The paper is organized as follows: The first section provides an introductory overview and outlines the research objectives. The second section offers a concise review of pertinent work published in reputable journals. The third section details the decision-making techniques and computational methodologies employed in this study. The fourth section presents the findings and insights derived from the numerical analysis of the research methodology. Finally, the fifth section offers a brief conclusion summarizing the research.

## 2. Literature review

Summarising and categorizing sports videos is evolving into a highly essential problem since it is critically necessary to instinctively recognize sports scenarios in order to improve sports investigation, officiating, training, and marketing. The categorization of sports records used a

lot of "black box" strategies, which are missing patterns that are simple for people to understand and examine. To provide clear, understandable results, fuzzy logic classification algorithms make use of language ideas and tags. However, the dimensionality curse of conventional fuzzy classification techniques may lead to substantial rule sets. Song and Hagras [6] have modified the guidelines of a fuzzy logic-based classification framework for categorizing sports clips utilizing the Big Bang-Big Crunch method, resulting in exceptional classification precision with fewer constraints and enhanced system interpretation. Numerous investigations on football videos have shown that the system, which uses only eight criteria and a typical rating precision of 83%, beats older black box classification techniques like neural networks. Elite talent is becoming increasingly difficult to locate as eSports gain in the mainstream. There is overwhelming rivalry since there are many billions of dollars in rewards available for this talent. To advance, athletes will go to enormous lengths to enhance their results. In addition to vigorous exercise and performance-enhancing medications, sportsmen search for smart mathematical instruments that may offer them helpful information on their abilities and limits. The proposed study [7] focuses on a fuzzy system that employs actual data to estimate an athlete's likelihood of victory in combat in the well-known electronic first-person fighting game Counter-Strike Global Offensive.

In the past few years, there has been a tremendous advancement in the computerization of sports video summaries. The bulk of methods used to recognize sports footage used SVMs and neural networks, which fail to offer frameworks that are simple for people to study and understand. Video sceneries may be considered a constant series of pictures, but due to the active structure of the video clip and the modifications brought on by difficult-to-distinguish components of the scene, the categorization issue is considerably more challenging than it is for a single picture. The proposed study [8] uses Interval Type-2 Fuzzy Logic categorization systems, the variables of which are altered utilizing the Big Bang-Big Crunch technique, to cope with the substantial amount of ambiguity in video clip categorization. This method provides instantaneous fashion scene categorization using the best criteria for broadcasting football game videos. By enabling an acceptable degree of classification precision to be obtained with a minimal number of guidelines, the suggested approach streamlines the system. Socially assistive robots (SARs) are being employed more often in dementia and eldercare settings. SARs must be customized for every single person to provide the appropriate support and encourage independent thought. Fuzzy logic systems with rules have been successful instruments for autonomous SAR decision-making. Even though this procedure can be time-consuming and expensive, fuzzy logic systems can be developed and changed to compensate for the ever-changing requirements, desires, and health issues of patients. Dell'Anna and Jamshidnejad [9] discussed EFS4SAR, a state-of-the-art framework for developing fuzzy logic that enables the autonomous synthesis of fuzzy regulations governing SAR activity. EFS4SAR integrates adaptive methods with conventional rule-based fuzzy logic structures. These algorithms mimic natural selection and have been proven to encourage innovative action. The suggested study uses mathematical models with real and simulated data to assess EFS4SAR. The findings indicate that individuals' particular values and health concerns, as well as the standards for uniqueness and efficacy in SAR innovation, are all served by the time-evolved fuzzy rules. EFS4SAR is more creative and performs better than earlier advances in fuzzy systems.

The concept of student-centered learning is examined in the suggested study. Enhancing student learning is the objective of a self-adapting web-based learning structure. A learner framework is used to determine the actions of an evolving system. The selection shows the instruction that, based on the learner's description, best explains a certain subject. To instruct on a subject, several teaching strategies are used. The learner framework then investigates the learner's psychological propensities to unearth subjective components including intellectual,

character, and learning preferences. It also emphasizes the advantages of various lecture formats for learning. Additionally, the SM develops a cognitive map (CM) periodically to establish unclear causal relationships between the different components of the presentation and the characteristics of the students. The SM forecasts the prejudice that a lecture's selection will have on the pupil's internship using a fuzzy-causal engine. An investigation and the computer prototype of the SM were used to test the conceptual, theoretical, and formal foundations of the method. According to the study, the experimental cohort of participants who employed this strategy learned 17% more on average than a comparable control group of participants who listened to lectures picked at random from another set of identical lectures [10]. Yaşlı and Yüncü [11] provided a framework for evaluating Turkish undergraduate culinary programmes. The Data Envelopment Analysis technique and fuzzy logic are used to measure the pedagogical outcomes of the curricular modules. During the assessment procedure, viewpoints that reflected the learning objectives of the culinary section were taken into account. The outcomes demonstrate that multiple programs serve the multiple philosophies of culinary learning in different ways, that particular courses may be eliminated from the syllabus, and that the status of other subjects has to be altered. Comparing the assumption to other training programmes that cover the same topic shows that it is accurate.

A conditional fuzzy inference (CF) technique for prediction is investigated in the study proposed by Hassanniakalager et al. [12]. Although fuzzy rules (FRs) can be removed, the suggested method has a number of drawbacks. The weak standards are eliminated by this conditional rule preference, leaving just the best rules for forecasting. This strategy could improve the usability of the underpinning system while simultaneously improving prediction precision. Numerous prediction activities that make use of market and football game data employ the CF idea. Relevant Standard algorithms such as SVM frameworks, Dynamic Neuro-Fuzzy Inference Systems, Decision Trees, and Vector Machines are used to assess the system's efficacy. The outcomes illustrate the CF's quantitative superiority over its norms. Rule-based logic, Bayesian inference, and within-game time data are all combined in the method proposed by [13] for predicting sports outcomes. Three key elements make up the structure's uniqueness. The rule-based reasoner and the Bayesian network component are the two main parts of the system. The outcomes of sporting contests may be predicted using either of the two distinct techniques. The concept that a team's goals may be roughly expressed by observable logical principles served as its inspiration despite the great level of stochasticity in athletic events. The recommended method can be applied to forecast the findings of games between teams that haven't previously played one another. Machine learning systems encounter a variety of issues as a result of these data restrictions. The suggested structure is capable of integrating a range of criteria for predicting the outcomes of athletic events, in contrast to most past research, which only included one variable, which was often the score. Compared to the vast bulk of prior research on the topic, the proposed study employ a knowledge-driven in-game time-series approach to forecast sporting activities. The recommended estimations are unquestionably more precise and reasonable since the system can depict the fluctuating phases of a sporting event in this way.

Techniques having naturalistic foundations can be found in computing intelligence, a subfield of AI. These techniques are all capable of shared intelligence and self-adaptation. These techniques' effectiveness and ease of use have made them useful for tackling issues in the social and natural sciences. The proposed study [14] has shown the potential advantages of using natural techniques in sports, especially when designing efficient, secure, and performance-specific training regimens. The advantages and potential uses of intelligent systems in sports are examined in this essay, along with any difficulties that could develop as the field of research progresses. With the development of AI, researchers can now create very accurate prediction

systems. Due to its efficiency, ML is employed in almost every industry. Football game estimations are a specific application of forecasting technology. Baboota and Kaur [15] have provided an overview of their ideas for developing a thorough statistical framework to forecast outcomes for the English Premier League. The set of characteristics necessary for finding the most crucial variables in forecasting the outcomes of a football game is constructed in the proposed study using feature modeling and interpretive statistical analysis. The next phase involves employing ML to build an extremely precise forecasting system. The results show how crucial some characteristics are to the effectiveness of the suggested models.

Fuzzy logic, a kind of human-like thinking, is increasingly used in sophisticated safety-relevant systems, choice-making, computer vision, control systems, and other data science uses. This is because of the fact that it creates database frameworks that are nonetheless adaptive and comprehensible while managing a variety of area and data ambiguity sources encountered in the actual world. Instead of using intricate AI algorithms and time-consuming testing processes, fuzzy logic may be utilized to perform surveillance tasks efficiently. The suggested article [16] educates readers on how fuzzy logic is used in BVD monitoring. Fuzzy logic-based approaches to the interpretation of visual and sensory data are thoroughly reviewed in the proposed study. The overview discusses the benefits, drawbacks, and difficulties of the current fuzzy logic-based video processing techniques for monitoring operations. In addition to identifying and assessing the datasets utilized by various methodologies, the study explores potential directions for future research based on an in-depth analysis of the work that has already been done in this fascinating field. Recently, there has been more ferocious competition in cricket. Because of their special bowling skills and modifications, Rahman et al. [17] have told how novice bowlers are progressing. A great bowler may totally confuse the hitter with a diversity of bowling strokes in a single over. Even though diverse bowling approaches may be perplexing to batters, the hold of the bowler might indicate a lot about what the bowler intends to bowl. To forecast the exact kind of ball the bowler intends to launch, the suggested research makes use of sophisticated ML, fuzzy logic, and deep learning frameworks. A grip collection containing 5573 photos of grips from 13 distinct delivery types is used in the study. Fuzzy logic centered on the L-channel of the CIE 1976 L*a*b* color space (CIELAB), which is derived from RGB, is used to illustrate a method for enhancing visual contrast. After training the recommended deep convolution neural network, VGG 16, KNN, and Decision Tree, the accuracy statistics were rather good.

It might be quite challenging and surprising to identify sequences in recorded movies. Patterns of activities happening continuously must be investigated in a variety of application areas, including video event movement recognition, in order to highlight the significant events from the video clips. Most modern responsive sequence classification techniques use the Gaussian mixture model (GMM) and dynamic time warping (DTW) to compare two spatial patterns, which might change at different rates. Video event identification algorithms must be developed to classify important occurrences in lengthy video clips. The suggested study offers a way for meaningful event extraction from the massive quantity of data in football media using DTW and Interval Type-2 Fuzzy Logic Networks [18]. The typical everyday assessment of stroke volume outside of a hospital in the lack of specific tools offers a challenge for an in-depth review of heart efficiency, even though the vitality of stroke volume estimation in comprehending the function of the circulatory system in individuals. Ahmedov and Amirjanov [19] set out to create a brand-new, effortless method for detecting stroke volume that utilizes the correlation between blood flow and gradually warming skin. The research's 92 individuals were chosen at random. By combining the Peltier effect with a thermoelectric cooler positioned on the wrist over artery pulsation, it was possible to calculate the duration of skin warming. The data comparison revealed a strong correlation between the generic-fuzzy logic

system's input and output characteristics. The study's technique provides a straightforward, transportable, and affordable substitute that may be employed to calculate the stroke volume at home. The recommended approach relies on a historical correlation between skin temperature and blood circulation. Before determining the individual's heart rate, the following device determines their age and changes in their pulse rate. It is now feasible to rely on the suggested approach for determining the circulatory system due to the adoption of the genetic-fuzzy framework, which also greatly enhanced the precision of measuring stroke volume. The medical expert can spot atrial disorders before they worsen and keep an eye on any heart abnormalities brought on by stress, including that experienced during physical exercise or competition.

Finding a suitable option set that can be statistically handled, represent the analyst's attitudes, and be incorporated into the selection model description may be difficult when creating individual decision structures. By providing a precise sampling strategy for the emergence of possible collections, the proposed study [20] would enhance the understanding of the decision-making process for choosing a recreational location. The proposed two-stage procedure begins with the creation of a concealed option set using a fuzzy logic system that includes rules. In the second phase, the set is included in a discrete option framework to predict decision-making attitude. It demonstrates the failure of the sampling strategies employed to select the standard set's preferred set from among its many alternatives. The suggested two-phase technique is evaluated by an everyday travel poll from Victoria, Australia, and nine distinct amusement selection patterns were created. Comparing these frameworks to the typical modeling paradigm reveals that they have exceptional accuracy and predictive ability. A great sports team is built by trainers utilizing their knowledge and evaluating the data gathered during practices and games. Fuzzy logic helps innovate decision-making paradigms due to the uncertainty of the facts and information. Palacios et al. [21] created a framework that uses a Genetic Fuzzy System to forecast a player's potential for success using a mix of domain expertise and previous data. The everyday life GFSs, whose research is now crucial to machine learning, are combined with a range of data forms in the resulting framework, which makes it fascinating. Players' assessments of their errors, comparisons of various metrics supplied by diverse viewers, and interval-valued instructional data are a few examples of this type of data. The suggested study showed that a GFS built on these final concepts exceeded the findings of the procedure's initial construction by using a modified principle-based logic approach and a perspective depiction of diverse types of knowledge.

The method advised by the research is to determine the amount of risk currently connected to sporting events based on specific criteria. These surveillance systems cannot give data that are meaningful and increase system security without a patient-particular appraisal. Receiving the right reaction as soon as possible is similarly vital in order to avoid unfavorable outcomes. To address the issue, the study proposed by Tóth-Laufer [22] provides a range of analysis structure mitigation methods that minimize the computing burden of risk analysis. These systems are extensions of the classic Mamdani-type reasoning approach that keep the system's beneficial features while simplifying computation and delivering outcomes on track with the classic system. Recent years have seen a rise in research in and utilization of augmented reality (AR). The bulk of these uses, nevertheless, are restricted to little indoor areas. Although AR has a number of extensive uses and contexts that could grow substantially from its utilization, these contexts haven't always been the primary focus of AR apps. Zollmann et al. [23] have investigated how AR could improve fans' enjoyment of live sporting events. The proposed study looks into the feasibility of live AR viewer encounters as well as the obstacles of incorporating AR in a large venue. A method for deploying AR apps across large areas was also investigated and a prototype was included. The work that is suggested by Etzion et al. [24] has introduced the event model (EM), which may be used to create, build, deploy, and manage

event-based systems. The model's event logic is presented in Excel-like tables in an attractive yet precise manner so that those without IT experience may grasp it. The EM employs the model-driven design technique and is a CIM (Computation Independent Model). Model-driven design and autonomous change are the goals. A method for transforming the CIM into practical event-driven software is offered in the study proposal. The proposed study illustrates the recommended technique utilizing a fraud identification scenario for mobile phones. Identifying sports outcomes has always been difficult due to the complex nature of sports, the performance of players, and the influence of numerous factors such as team positions with different strategies, player statistics, and historical data. Accurate predictions can significantly impact decision-making processes in sports management, betting, and team planning [25]. With the advancement of AI, an approach for enhancing predictive performance in sports [17]. With this advancement, fuzzy logic-based models have gained attention for their ability to handle uncertainty and provide more adaptable predictions in complex environments [26]. Combining these models with advanced multi-criteria decision-making techniques like CRITIC (Criteria Importance Through Intercriteria Correlation) and VIKOR (VIšeKriteri-jumska Optimizacija I Kompromisno Resenje) further enhances their predictive power [27].

## 3. Methodology

Fuzzy logic-based frameworks gained widespread attention in the past few years and are widely utilized in the sports industry for match outcomes predictions. Specifically, fuzzy logic-based models are used to forecast the results of sporting events for a variety of reasons. Increasing the precision and dependability of result forecasts is one of the main goals. Fuzzy logic-based models may produce forecasts that take into account uncertainty and take into account the complexity of sporting events by taking into consideration a wide variety of parameters such as team performance, player statistics, historical data, and other pertinent criteria. The key aim of this research is to evaluate the applications of fuzzy logic-based models in predicting outcomes of sports events that may help sports analysts in the selection of a suitable alternative from a set of different options. The available work has been thoroughly reviewed to choose some common features and alternatives from them for comprehensive evaluation. Then, we utilized hybrid techniques known as the CRITIC-VIKOR method which aims to determine the inter-criteria correlation between criteria by measuring their weights and then ranking the alternative to identify a suitable option. The CRITIC method aims to weighted the criteria and identify their weight, while the VIKOR serves to rank the alternatives by proposing a compromise solution. Finally, all the models were ranked based on the S, R, and $Q_i$ scores and a suitable alternative was determined. Specific sports perform poorly when predicted athletic events are made using renowned artificial neural network-based technologies. The main issue in this scenario is the usage of diverse forms of data and designs, including arithmetic figurative, and interval data as data sources. Krutikov et al. [27] describe a technique that involves cascading multiple kinds of neural networks. Every component generates a different training batch depending on the model kind and cascade design used. Components in lower-level tiers employ data analysis outputs from every level as collections of data. The gadget was first controlled via the MATLAB software suite. The outcomes of the tests undertaken validate the efficiency of cascade components. By utilizing Quartus CAD and an Altera Cyclone III FPGA to build a series of neural networks, the system throughput is substantially enhanced. For instance, this technique may be used in betting establishments. Interest in multi-energy micro-grids as a novel power source alternative has increased as a result of the current energy crisis and the escalating environmental issue. An in-depth, scientific investigation of the multi-energy microgrid is therefore very helpful and might offer financial advice. The proposed

research [28] suggests a microgrid assessment approach using prospect theory and the VIKOR approach in light of the stated context. The first stage is to develop a thorough index system that evaluates microgrids while taking into account elements like economics, energy efficiency, and environmental considerations. Prospect theory is used to create the prospect value matrix while taking into account the analyst's arbitrary views. Both the CRITIC technique and the augmented analytic hierarchy process (AHP) approach are used to calculate the overall weight. Using the VIKOR technique, more criteria are added to conduct a complete review. To illustrate how accurate and practical the suggested approach is, the last step includes an example analysis. Selecting a network may be difficult, particularly in inconsistent situations, due to the large diversity of access modes and equipment, as well as the varying QoS needs of the relevant apps and offerings. The AHP, the CRITIC technique, the VIKOR method, and the Total Order Preference by Similarity to the Ideal Solution (TOPSIS) approach are some of the MADM procedures that Sgora et al. [29] have assessed for effectiveness. The statistical findings demonstrate that the Always Best Connection (ABC) idea is guaranteed by a blend of the currently in use approaches. Furthermore, the Step-wised description of the proposed algorthim are,

Step 1: Collect a comprehensive dataset that includes historical data on sports events, team and player performance metrics, and other influencing factors.

Step 2: Preprocess the data to ensure quality, consistency, and accuracy, including handling missing values, normalizing data, and preparing the evaluation matrix for analysis.

Step 3: Determine the criteria for evaluating the predictive models, such as interpretability, historical data reliability, statistical modeling accuracy, and other relevant metrics.

Step 4: Apply the CRITIC Method for Criteria Weighting

4.1 Construct the Evaluation Matrix: Develop a matrix where rows represent the alternatives (predictive models) and columns represent the criteria.

4.2 Normalize the Matrix: Standardize the data to make the criteria comparable, scaling the values between 0 and 1.

4.3 Calculate Correlation Coefficients: Determine the correlation between each pair of criteria to understand their interrelationships.

4.4 Measure the Conflict Between Criteria: Use the correlation data to calculate the measure of conflict for each criterion, which reflects how criteria relate.

4.5 Calculate Criteria Weights: Determine the importance of each criterion using the standard deviation and measure of conflict to assign weights that will be used in the VIKOR method.

Step 5: Rank the Alternatives Using the VIKOR Method

5.1 Construct the Normalized Decision Matrix: Use the normalized values and the weights obtained from the CRITIC method.

5.2 Calculate the Utility (S) and Regret (R) Measures: Compute the utility measure for each model to assess overall performance and the regret measure to evaluate the worst performance of each criterion.

5.3 Calculate the VIKOR Index (Q): Combine the S and R measures to find the VIKOR index for each model, which identifies the compromise solution by balancing overall performance and worst-case scenarios.

5.4 Rank the Models: Rank the alternatives based on their Q values, with the model having the lowest Q value considered the most suitable.

Step 6: Selection of the Optimal Model. Based on the VIKOR ranking, select the model that offers the best balance of utility and regret, effectively addressing the uncertainties and complexities of sports event prediction. Validate the chosen model's performance by comparing it against other alternatives using key metrics.

## 3.1 Research framework

A thorough and precise assessment is imperative for decision-makers and sports experts to make effective choices and select an appropriate model for predicting outcomes in sports events. To enhance the sports industry, this research employs a hybrid model comprising the CRITIC-VIKOR method for an in-depth exploration of fuzzy logic-based models in predicting sports event outcomes. The study has been structured into distinct phases to ensure precise evaluation and achieve its research objectives. Initially, an extensive review of relevant articles published in prominent journals, including IEEE, Wiley, ScienceDirect, ACM, and Springer, was conducted to extract comparable and essential features. These features were assessed for their relative importance, and choices were ranked based on their respective weights. Next, the CRITIC method was applied to assign weights to the criteria and calculate their values. In the final stage, the VIKOR method was employed to rank all the alternatives and determine the most suitable choice. This study successfully accomplishes its goal of evaluating the effectiveness of the fuzzy logic-based model in predicting sports event outcomes. Fig 1 illustrates the sequential flow of this research.

## 3.2 Features extraction and selection

The unique features of a literary work that have been encountered or extracted are made obvious when we investigate imaginative traits that can be acknowledged. Numerous variables

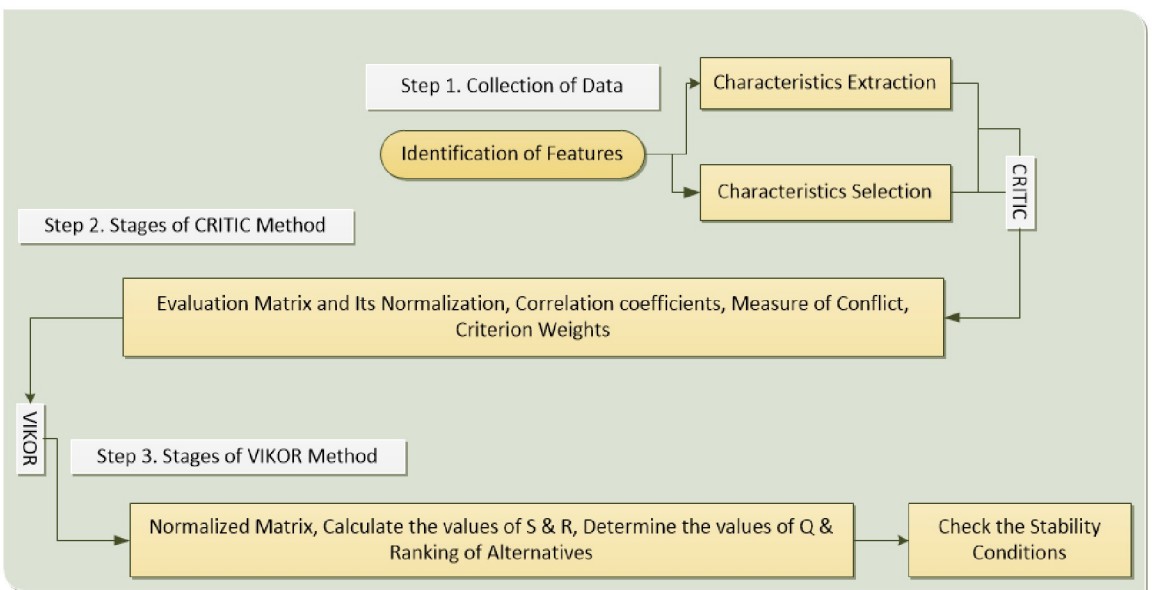

**Fig 1. Research route of this study.**

**Table 1. Extracted characteristics.**

| Authors | Characteristics | Citation No. |
|---|---|---|
| Song and Hagras | Summarization, classification, sports scenes, refereeing, training, linguistic rules, accuracy, optimization, interpretability, streaming media | [6] |
| Deane and Khuman | Talent intensifies, fierce competition, intensity, team performance, player strengths, real-world data, enhancement, fuzzy logic, player statistics | [7] |
| Song and Hagras | Significant, automation, summarization, image classification, dynamic nature, occlusions, sports analytics, uncertainties, real-time scene, classification accuracy, interpretability | [8] |
| Dell'Anna and Jamshidnejad | Effective, assistance, divergent thinking, creative behaviors, automated decision, modification, synthetic data, originality, creativity, personal preferences | [9] |
| Peña-Ayala et al. | Self-adaptation, complexity level, psychological attributes, cognitive preferences, dynamic nature, team performance, synthetic data | [10] |
| Yaşlı and Yüncü | Performance, perspectives, effectiveness, transformation, player competencies, knowledge, satisfactory maturity, flexibility, fuzzy logic | [11] |
| Hassanniakalager et al. | Forecasting performance, interpretability, match datasets, statistical accuracy, benchmarks, data uncertainties, transparency, generalization value, outcome prediction, manageable level, decision process | [12] |
| Min et al. | Sports prediction, stochastic, statistical data, scanty, outcomes prediction, realistic, reasonable predictions, stability, complexity level, availability | [13] |
| Fister et al. | Computational intelligence, adaptability, efficiency, simplicity, effective training, team performance, data uncertainties, reasoning process, knowledge, player statistics | [14] |
| Baboota and Kaur | Predictive precision, effectiveness, outcomes prediction, performances, probability score, data mining, economic value, error rate, complexity level, profitability, intensity, availability, computer vision | [15] |
| Muhammad et al. | Active practice, activity recognition, intelligent surveillance, computational complexity, purposeful usage, activity monitoring, actionable decisions, human-like logic, image processing | [16] |
| Rahman et al. | Competitive sports, uniqueness, bowling styles, image classification, image enhancement, decision tree, predictive precision, historical data | [17] |
| Song and Hagras | Sequence classification, complexity level, data uncertainty, summarization, adaptive sequences, activity detection, dynamic programming, dynamic time wrapping | [18] |
| Ahmedov and Amirjanov | Readings, comprehensive analysis, performance, non-invasive technique, variability, fuzzy logic, dependency, portable solution, accuracy | [19] |
| Hassan et al. | Feasibility, computationally manageable, modeling specifications, fuzzy logic, prediction potential, explanatory variables, predictive ability, sensitivity, data specifications, impressive outcomes | [20] |
| Palacios et al. | Data collection, knowledge, team performance, subjective perceptions, training data, possibilistic representation, original formulation, decision model, historical data | [21] |
| Tóth-Laufer | Risk level, sports activity, monitoring system, realistic results, security, computational need, modifications, computational complexity, adaptation models, statistical modeling, fuzzy logic | [22] |
| Zollmann et al. | Augmented reality, suitability, conception, design, predictive ability, statistical modeling, computational complexity, stochastic | [23] |
| Etzion et al. | Event-driven application, maintenance, model accessibility, designing approach, automatic transformation, fraud detection, IT skills | [24] |

have an impact on these traits. Table 1 displays the fundamental factors that were built using the data offered.

The comparable qualities relevant to making decisions originated after diligently researching different articles, and they are depicted in Fig 2.

### 3.3 CRITIC and VIKOR technique

In multi-criteria decision-making processes, the CRITIC technique is put to use to establish the weightings or relevance of several parameters. The CRITIC technique involves the decision-maker's values through pairwise analyses to provide an organized and logical way of the selection process. Setting criterion priority and assessing alternatives according to how they perform against certain criteria are both aided by this. On the other hand, the VIKOR

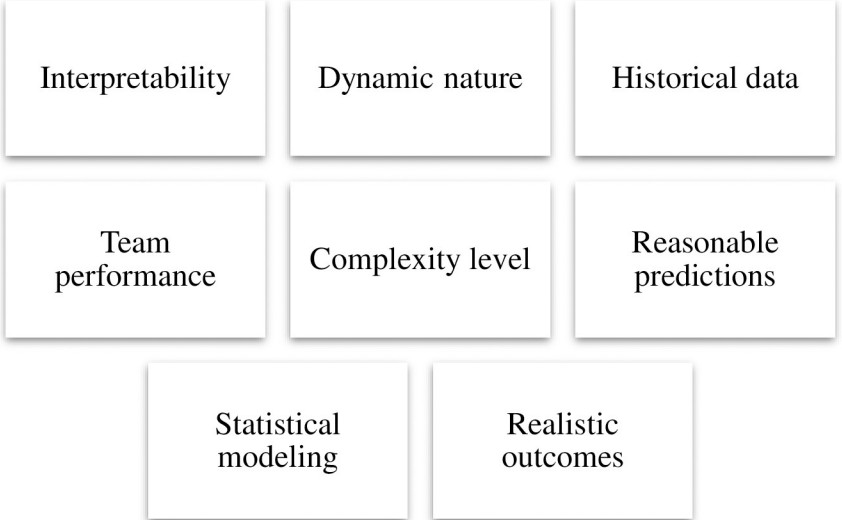

**Fig 2. Selected criteria.**

technique allows for the comparison of alternatives to establish benchmarks and gauge how efficiently they behave. It presents a compromise solution by looking into both the finest and least good efficiency of alternatives. It may further help to take into account all the different factors and their relative weights to arrive at a compromise solution. When it's important to assess several features before contemplating the taken alternatives, such as when choosing a project, predicting outcomes in sports events, evaluating suppliers, or making strategic decisions, the technique may be applied in a range of situations. The proposed methods follow a systematic way with the following steps:

In CRITIC,

**Step 1.** Develop an evaluation matrix: An evaluation matrix has been constructed containing options, criteria, and numerical values assigned to them. Let X indicate the evaluation matrix with $n$ rows (alternatives) and $m$ columns (criteria);

$$\mathrm{X} = \left[\mathrm{X_{ij}}\right] = \begin{array}{c} \\ A_1 \\ . \\ . \\ . \\ . \\ . \\ A_n \end{array} \begin{bmatrix} C_1 & \cdots & \cdots & C_n \\ X_{11} & X_{12} & \ldots & X_{1n} \\ X_{21} & X_{22} & \ldots & X_{2n} \\ X_{31} & X_{32} & \ldots & X_{3n} \\ \vdots & \cdots & \cdots & \vdots \\ X_{m1} & X_{m2} & \ldots & X_{mn} \end{bmatrix} \qquad (1)$$

**Step 2.** Calculation of ideal best and worst criterion outputs: The best and worst factors outputs have been computed for each criterion. For beneficial criteria, the values are calculated

as;

$$X_j^b = max_{i=1}^n X_{ij}, X_j^w = min_{i=1}^n min_{i=1}^n \quad (2)$$

If the factors have been non-beneficial, then the outcomes are computed as;

$$X_j^b = min_{i=1}^n X_{ij}, X_j^w = max_{i=1}^n X_{ij} \quad (3)$$

**Step 3.** Normalized Matrix: In this step, the evaluation matrix has been normalized and the elements of the normalized matrix are computed as;

$$\hat{R}_{ij} = \frac{X_{ij} - min\left(X_{ij}\right)}{max\left(X_{ij}\right) - min\left(X_{ij}\right)} \quad (4)$$

**Step 4.** Correlation Coefficient: The correlation coefficient between pairs of criteria has been measured and outputs are calculated as;

$$\rho_{jk} = \frac{\sum_{i=1}^m \left(r_{ij} - \acute{r}_j\right)(r_{ik} - \acute{r}_k)}{\sqrt{\left(\sum_{i=1}^m \left(r_{ij} - \acute{r}_j\right) 2 \sum_{i=1}^m (r_{ik} - \acute{r}_k)\right)^2}} \quad (5)$$

**Step 5.** Measure of Conflict (MoC): Eq (6) has been implemented for the measurement of the MoC for each criterion.

$$\text{Measure of conflict (MoC)} = \left(\sum_{jt=1}^n \left(1 - r_{jj}'\right)\right) \quad (6)$$

**Step 6.** Criterion Weights: Finally, the relative importance of each criterion has been obtained and criterion weights are calculated as;

$$\text{Weights}\left(W_j\right) = \frac{C_j}{\sum_{j=1}^n C_j} \quad (7)$$

In VIKOR,

**Step 7.** Draw a normalized matrix: Eq (8) is implemented into the matrix, which then undergoes a normalization process by making the values of the evaluation matrix decision comparable.

$$\textit{Normalized Matrix}\left(N_{ij}\right) = \frac{X_j^+ - X_{ij}}{X_i^+ - X_i^-} \quad (8)$$

**Step 8.** Determination of S and R: The values of "S" and "R" for every alternative has been calculated by using Eqs (9) and (10), respectively;

$$S_i = \sum_{j=1}^{m} \left( W_j * N_{ij} \right) \tag{9}$$

$$R_i = max_j \left( W_j * N_{ij} \right) \tag{10}$$

**Step 9.** The required values of $S^+$, $S^-$, $R^+$, and $R^-$ have been calculated by utilizing the Eqs (11) to (14), correspondingly;

$$S^+ = min_i(S_i) \tag{11}$$

$$S^- = max_i(S_i) \tag{12}$$

$$R^+ = min_i(R_i) \tag{13}$$

$$R^- = max_i(R_i) \tag{14}$$

**Step 10.** Measurement of $Q_i$ scores: These scores have been obtained by utilizing Eq (15);

$$Q_i = \mathcal{V} * \frac{S_i - S^*}{S^- - S^*} + (1 - \mathcal{V}) * \frac{R_i - R^*}{R^- - R^*} \tag{15}$$

**Step 11.** The S, R, and Qi outcomes can be sorted in a ranking that ascends to find the alternative within a set of analyzed possibilities. A favorite one has a lower $Q_i$ outcome while fulfilling one of the final two stabilization conditions, whereas the unlikely choice has a higher $Q_i$ yields.

C-1: Q(A2)−Q(A1) ≥ DQ where DQ = 1/J−1

C-2: An alternative that ranked 1st should also have a higher score in means of S and/or R.

The people making decisions or researchers need to be made aware that a compromise grading has been assigned if the stability requirements have not been achieved.

Numerical Work and Results of the CRITIC and VIKOR Techniques.

First, we outline the decision problem in detail and list the factors that were taken into account when reaching the choice. The key objective of this numerical work is to assess the applications of Fuzzy logic-based models in predicting outcomes of sports events. The CRITIC method applies to weight the criteria, while the VIKOR method aims to rank the alternative and proposed a compromised solution. For this purpose, different fuzzy logic-based models including Models 1 through 5 are available as choices, and eight criteria have been discovered and stated, including interpretability, dynamic nature, historical data, team performance, complexity level, reasonable predictions, statistical modeling, and realistic outcomes. The entire set of selected criteria in this study is beneficial in nature. The calculation work and their results are as follows;

In CRITIC,

Using Eq (1), an evaluation matrix has been created that enables comparisons between each criterion and every other criterion. Additionally, we analyze the relative significance or the priority of one factor over another to give mathematical scores to signify the degree of preference. The Saaty scale [30], which spans from 1 (equal significance) to 9 (high importance), is a popular measure used by experts for pairwise comparisons. The options, criteria, and data that are going to be utilized in the assessment are employed for generating an evaluation matrix in this stage and the resultant matrix is as shown in Table 2.

For the sake of making the evaluation matrix values comparable, the matrix is transformed into a normalized matrix utilizing Eq (4). Table 3 shows the resulting matrix.

In this step, the inter-criteria correlation has been captured and their relationship between each other has been identified using Eq (5). Table 4 shows the generated matrix of calculated correlation coefficients.

**Table 2. Evaluation matrix.**

| Criteria / Alternatives | Interpretability | Dynamic nature | Historical data | Team performance | Complexity level | Reasonable predictions | Statistical modeling | Realistic outcomes |
|---|---|---|---|---|---|---|---|---|
| Model1 | 7 | 3 | 8 | 6 | 4 | 2 | 9 | 5 |
| Model2 | 2 | 6 | 5 | 4 | 7 | 3 | 5 | 4 |
| Model3 | 5 | 4 | 7 | 5 | 2 | 6 | 8 | 3 |
| Model4 | 8 | 5 | 4 | 9 | 6 | 5 | 3 | 7 |
| Model5 | 3 | 9 | 6 | 7 | 3 | 4 | 2 | 6 |
| $X_{max}$ | 8 | 9 | 8 | 9 | 7 | 6 | 9 | 7 |
| $X_{min}$ | 2 | 3 | 4 | 4 | 2 | 2 | 2 | 3 |

**Table 3. Normalized evaluation matrix.**

| | Interpretability | Dynamic nature | Historical data | Team performance | Complexity level | Reasonable predictions | Statistical modeling | Realistic outcomes |
|---|---|---|---|---|---|---|---|---|
| Model1 | 0.8333 | 0.000 | 1.000 | 0.400 | 0.400 | 0.000 | 1.000 | 0.500 |
| Model2 | 0.000 | 0.500 | 0.250 | 0.000 | 1.000 | 0.250 | 0.429 | 0.250 |
| Model3 | 0.500 | 0.167 | 0.750 | 0.200 | 0.000 | 1.000 | 0.857 | 0.000 |
| Model4 | 1.000 | 0.333 | 0.000 | 1.000 | 0.800 | 0.750 | 0.143 | 1.000 |
| Model5 | 0.167 | 1.000 | 0.500 | 0.600 | 0.200 | 0.500 | 0.000 | 0.750 |

**Table 4. Correlation coefficients between pairs of criteria.**

| | Interpretability | Dynamic nature | Historical data | Team performance | Complexity level | Reasonable predictions | Statistical modeling | Realistic outcomes |
|---|---|---|---|---|---|---|---|---|
| Interpretability | 1.000 | -0.639 | 0.062 | 0.663 | -0.047 | 0.124 | 0.257 | 0.434 |
| Dynamic nature | -0.639 | 1.000 | -0.412 | 0.147 | 0.010 | 0.069 | -0.847 | 0.343 |
| Historical data | 0.062 | -0.412 | 1.000 | -0.411 | -0.686 | -0.300 | 0.778 | -0.500 |
| Team performance | 0.663 | 0.147 | -0.411 | 1.000 | 0.038 | 0.247 | -0.528 | 0.904 |
| Complexity level | -0.047 | 0.010 | -0.686 | 0.038 | 1.000 | -0.381 | -0.308 | 0.305 |
| Reasonable predictions | 0.124 | 0.069 | -0.300 | 0.247 | -0.381 | 1.000 | -0.207 | -0.100 |
| Statistical modeling | 0.257 | -0.847 | 0.778 | -0.528 | -0.308 | -0.207 | 1.000 | -0.674 |
| Realistic outcomes | 0.434 | 0.343 | -0.500 | 0.904 | 0.305 | -0.100 | -0.674 | 1.000 |

**Table 5. Calculation of the measure of conflict.**

|  | Interpretability | Dynamic nature | Historical data | Team performance | Complexity level | Reasonable predictions | Statistical modeling | Realistic outcomes | MoC |
|---|---|---|---|---|---|---|---|---|---|
| Interpretability | 0.000 | 1.639 | 0.938 | 0.337 | 1.047 | 0.876 | 0.743 | 0.566 | 6.146 |
| Dynamic nature | 1.639 | 0.000 | 1.412 | 0.853 | 0.990 | 0.931 | 1.847 | 0.657 | 8.329 |
| Historical data | 0.938 | 1.412 | 0.000 | 1.411 | 1.686 | 1.300 | 0.222 | 1.500 | 8.470 |
| Team performance | 0.337 | 0.853 | 1.411 | 0.000 | 0.962 | 0.753 | 1.528 | 0.096 | 5.942 |
| Complexity level | 1.047 | 0.990 | 1.686 | 0.962 | 0.000 | 1.381 | 1.308 | 0.695 | 8.070 |
| Reasonable predictions | 0.876 | 0.931 | 1.300 | 0.753 | 1.381 | 0.000 | 1.207 | 1.100 | 7.549 |
| Statistical modeling | 0.743 | 1.847 | 0.222 | 1.528 | 1.308 | 1.207 | 0.000 | 1.674 | 8.531 |
| Realistic outcomes | 0.566 | 0.657 | 1.500 | 0.096 | 0.695 | 1.100 | 1.674 | 0.000 | 6.287 |

The measure of conflict has been captured for each criterion and its values have been obtained using Eq (6). Table 5 contains the generated matrix and calculated outcomes of the measure of conflict.

The required values including standard deviation and quantity of information have been determined before going towards the determination of criterion weights. After the calculation of these values, the criteria have been weighted to determine their relative significance using Eq (7). Table 6 presents different outcomes and the calculated weights of each criterion.

The weightage of each criterion has been determined using the CRITIC approach and is described in graphical form. Fig 3 shows the identified weights of the entire criteria.

In VIKOR,

In this step of the VIKOR, we normalized the evaluation matrix constructed for existing decision-making issues. Eq (8) was used to normalize the initial data set for each criterion and make them all have the same scale. This process guarantees that all criteria are similar and prevents bias brought on by disparate measuring scales or units. Table 7 shows the calculated outcomes, criterion weights, and generated normalized matrix of the VIKOR method.

In this step, the criterion weights determined by the CRITIC method are multiplied by the respective column in a normalized matrix to get the required values. Then, these values have been processed using Eqs (9) and (10) to obtain the S and R outcomes. Table 8 displays the identified values of S and R.

**Table 6. Criterion weights.**

| Criteria | Standard deviation $(\sigma_j) = \sqrt{\sum_{i=1}^{n} \frac{(\sigma_i - \bar{\sigma})^2}{n-1}}$ | Quantity of information $\left(C_j\right) = \sigma_j * \left(\sum_{j'=1}^{n} \left(1 - r_{jj'}\right)\right)$ | Weights |
|---|---|---|---|
| Interpretability | 0.425 | 2.612 | 0.1089 |
| Dynamic nature | 0.384 | 3.196 | 0.1333 |
| Historical data | 0.395 | 3.348 | 0.1396 |
| Team performance | 0.385 | 2.286 | 0.0953 |
| Complexity level | 0.415 | 3.347 | 0.1396 |
| Reasonable predictions | 0.395 | 2.984 | 0.1245 |
| Statistical modeling | 0.436 | 3.716 | 0.1550 |
| Realistic outcomes | 0.395 | 2.485 | 0.1037 |

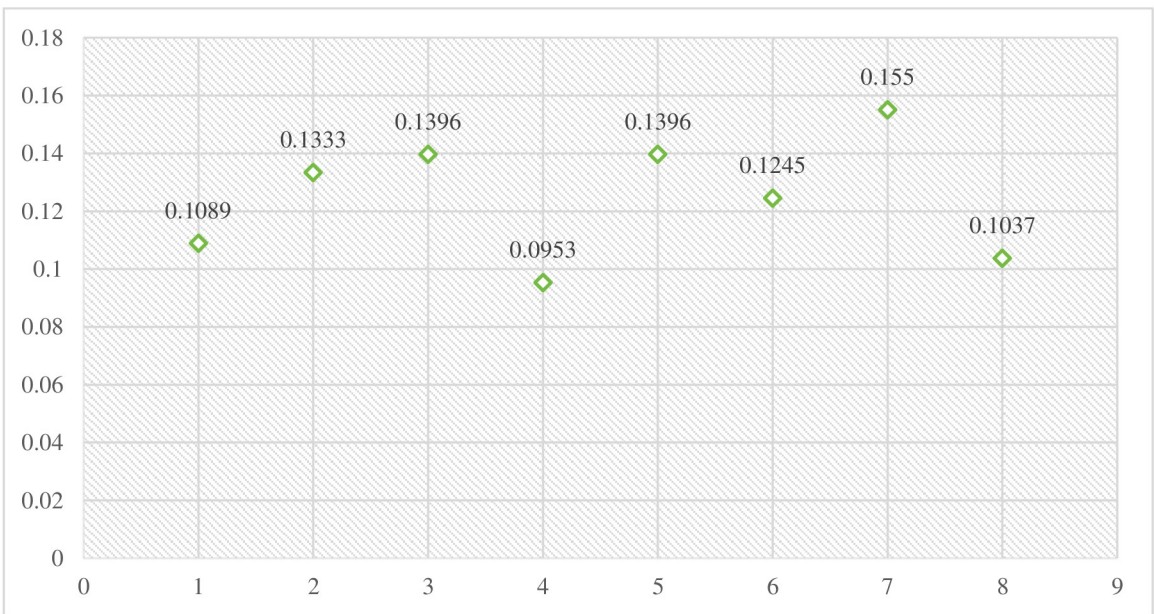

**Fig 3. Weightage of criteria.**

**Table 7. Normalized evaluation matrix of VIKOR.**

| Criteria / Alternatives | Interpretability | Dynamic nature | Historical data | Team performance | Complexity level | Reasonable predictions | Statistical modeling | Realistic outcomes |
|---|---|---|---|---|---|---|---|---|
| Model1 | 0.167 | 1.000 | 0.000 | 0.600 | 0.600 | 1.000 | 0.000 | 0.500 |
| Model2 | 1.000 | 0.500 | 0.750 | 1.000 | 0.000 | 0.750 | 0.571 | 0.750 |
| Model3 | 0.500 | 0.833 | 0.250 | 0.800 | 1.000 | 0.000 | 0.143 | 1.000 |
| Model4 | 0.000 | 0.667 | 1.000 | 0.000 | 0.200 | 0.250 | 0.857 | 0.000 |
| Model5 | 0.833 | 0.000 | 0.500 | 0.400 | 0.800 | 0.500 | 1.000 | 0.250 |
| Criteria Weights | 0.1089 | 0.1333 | 0.1396 | 0.0953 | 0.1396 | 0.1245 | 0.1550 | 0.1037 |

Eqs (11) to (14) have been used to calculate the required values before going towards the determination of $Q_i$ outcomes. After these values, the $Q_i$ score has been calculated using Eq (15) to rank the alternatives based on these values. The stability conditions were checked and a set of various alternatives were ranked based on their S, R, and Q values. Table 9 shows the entire calculation of $Q_i$ values and the ranking of alternatives.

**Table 8. S and R calculation.**

| | Interpretability | Dynamic nature | Historical data | Team performance | Complexity level | Reasonable predictions | Statistical modeling | Realistic outcomes | S | R |
|---|---|---|---|---|---|---|---|---|---|---|
| Model1 | 0.018 | 0.133 | 0.000 | 0.057 | 0.084 | 0.125 | 0.000 | 0.052 | 0.469 | 0.133 |
| Model2 | 0.109 | 0.067 | 0.105 | 0.095 | 0.000 | 0.093 | 0.089 | 0.078 | 0.635 | 0.109 |
| Model3 | 0.054 | 0.111 | 0.035 | 0.076 | 0.140 | 0.000 | 0.022 | 0.104 | 0.542 | 0.140 |
| Model4 | 0.000 | 0.089 | 0.140 | 0.000 | 0.028 | 0.031 | 0.133 | 0.000 | 0.420 | 0.140 |
| Model5 | 0.091 | 0.000 | 0.070 | 0.038 | 0.112 | 0.062 | 0.155 | 0.026 | 0.554 | 0.155 |

**Table 9. Calculation of Q values and ranking of alternatives.**

| Alternatives | S | R | $V * \frac{S_i - S^*}{S^- - S^*}$ | $(1 - V) * \frac{R_i - R^*}{R^- - R^*}$ | $Q_i$ | Ranking |
|---|---|---|---|---|---|---|
| Model1 | 0.469 | 0.133 | 0.113 | 0.265 | 0.377 | 2 |
| Model2 | 0.635 | 0.109 | 0.500 | 0.000 | 0.500 | 3 |
| Model3 | 0.542 | 0.140 | 0.283 | 0.333 | 0.616 | 4 |
| Model4 | 0.420 | 0.140 | 0.000 | 0.333 | 0.333 | 1 |
| Model5 | 0.554 | 0.155 | 0.310 | 0.500 | 0.810 | 5 |
| $S^+$, $R^+$ | 0.420 | 0.109 | | | | |
| $S^-$, $R^-$ | 0.635 | 0.155 | | | | |

The ranking of each alternative determined by the VIKOR method has been displayed in graphical form. Fig 4 shows the ranking of various alternatives.

## 4. Results and discussion

Making predictions regarding the outcomes of sporting events has long piqued the attention and excitement of both sports followers and professionals. Forecasting the outcomes of sporting events correctly enhances the entire viewer's pleasure and has implications for issues like sports betting, team strategies, and fan involvement. It has been demonstrated that fuzzy logic-based models, which are effective at managing data ambiguity and instabilities, are a good option for predicting athletic events. These simulations provide a thorough framework for assessing and analyzing the different aspects impacting the result of sporting events when paired with multi-criteria evaluation methods like CRITIC and VIKOR. Fuzzy logic offers a versatile and reliable statistical paradigm that takes into account the fuzziness and contradiction frequently present in sports data. Fuzzy logic enables the depiction of ambiguous or partially intersecting categories, in contrast to conventional binary techniques that only classify

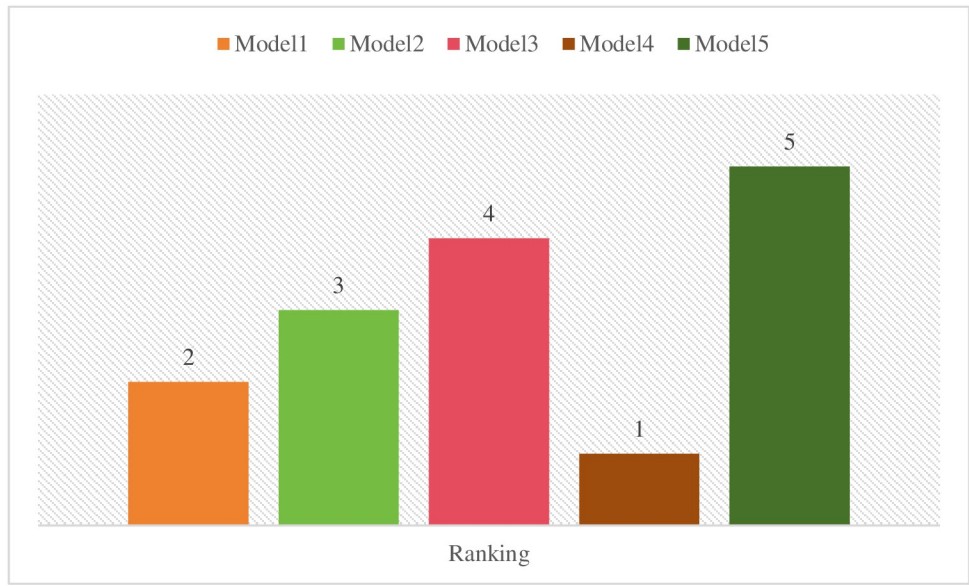

**Fig 4. Ranking of alternatives.**

results as a win or lose. This makes it possible to more accurately assess the possibility of various results, including having a significant probability of winning, an average probability of winning, or a small probability of winning, depending on a number of variables and their relative strengths of effect.

Fuzzy logic-based approaches utilize linguistic parameters and fuzzy rules to reflect the fuzziness and uncertainty that are typically present in sports data. In place of adopting exact simple results (win or lose), these models enable grades or levels of adherence associated with various results (e.g., the high likelihood of winning, the intermediate chance of winning, little likelihood of winning). Fuzzy logic is a suitable option for sports forecasting because the result is impacted by a range of factors because of its versatility. To overcome the evaluation issue and assess these models, the CRITIC and VIKOR methods were implemented in this research. The CRITIC approach has been applied to assess the relative weight given to various factors throughout a decision-making procedure. To determine the connections between various criteria and to determine the appropriate weights, the inter-criteria matrices are calculated. We also evaluated the importance of numerous characteristics in forecasting sports results by using CRITIC to fuzzy logic-based frameworks. On the other hand, VIKOR is a decision-support approach that seeks to reconcile many competing factors. It determines the optimum option based on a thorough assessment of the factors, accounting for both the greatest possible collective benefit and the least possible particular regret. We obtained the best results by incorporating VIKOR into fuzzy logic-based models and concurrently taking into account a number of different parameters.

The entire calculational work of the CRITIC-VIKOR methods indicates that the statistical modeling has the highest relative weight with a percentage of 16, by following historical data and complexity with a percentage of 14, and other relevant criteria with a different percentage that shows their relative importance throughout the evaluation process. Then, the VIKOR method was applied, which shows that the alternative Model4 has a top-performing model with a lower percentage of Q and 1st placed, by following the remaining fuzzy logic-based models with different percentages and ranking in the specific rule of decision-making technique. Although, the Model5 alternative secured the bottom place with higher Q percentage and known as least preferred choice by experts in predicting sports events outcomes. These findings explore how the CRITIC and VIKOR approaches might be used to apply fuzzy logic-based frameworks to the prediction of sporting events. We go into further detail about this strategy's efficacy, possible benefits, and difficulties in using it in the complex and uncertain realm of sports. We can learn a lot about these models' prediction powers and their significance for sports analytics by assessing the combination of fuzzy logic, CRITIC, and VIKOR.

The precision and dependability of sports outcome forecasts can be improved by combining fuzzy logic-based models with the CRITIC and VIKOR approaches. These techniques provide a more thorough examination by taking into account various parameters and their corresponding relevance weights. Furthermore, they help decision-makers by allowing them to weigh many goals and create trade-offs between various criteria. Sports events may nevertheless be unexpected by nature because of the intricate relationships that exist between players, teams, and other outside variables. Although fuzzy logic and the use of the CRITIC and VIKOR approaches can enhance forecasts, they cannot provide absolute precision. The result of a game might nevertheless be dramatically impacted by unanticipated additional factors or uncontrollable events that occur during it. The efficiency of these techniques is also influenced by the fuzzy logic model's architecture, the choice of the appropriate criteria weights using CRITIC, and the accuracy and applicability of the information being provided. It is essential to regularly make improvements to these models using the most recent information in order to retain their applicability and predictability in predicting sports results. Finally, using the

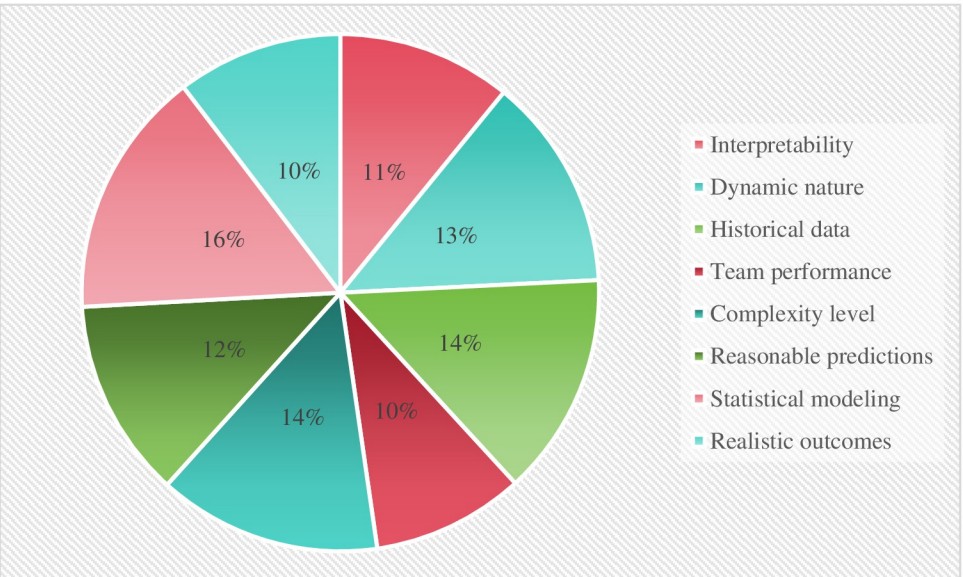

**Fig 5. Weights of criteria.**

CRITIC and VIKOR approaches to evaluate the use of fuzzy logic-based models for predicting sporting events gives a methodical and complete solution. These techniques may manage ambiguity and numerous factors, producing forecasts with greater accuracy. But how well they work depends on the nature of sporting events, the accuracy of the data, and how well the models are created and maintained. Figs 5 and 6 indicate the comprehensive results of this research including the weightage of each criterion and the ranking of alternatives.

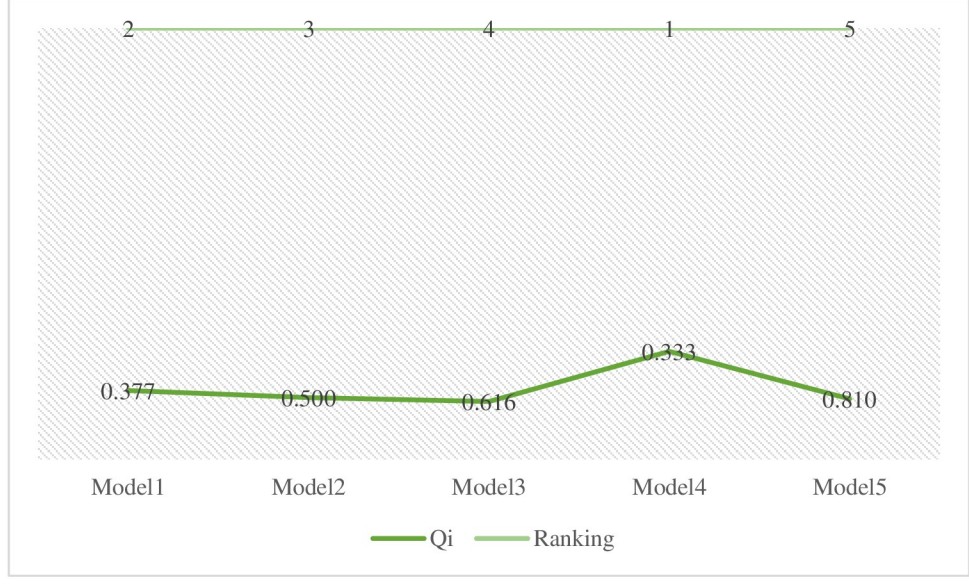

**Fig 6. $Q_i$ Values and ranking of alternatives.**

Fig 5 contains the comprehensive weight of each criterion that was determined by the CRITIC approach.

Fig 6 represents the comprehensive values of $Q_i$ and the ranking of each alternative identified by the VIKOR method.

## 5. Conclusion

Recently, fuzzy logic-based models have gained attention for predicting sports outcomes, especially in sports forecasting where precise numerical values are difficult to obtain. Accurately predicting results in sports remains a significant challengeIn this research, fuzzy logic-based modeling and the CRITIC-VIKOR method are proposed approaches to addressing the inherent ambiguities and complexities in sports forecasting. The CRITIC method facilitates systematic parameter selection by identifying and prioritizing relevant factors. Meanwhile, the VIKOR method streamlines the selection of appropriate fuzzy logic-based models by considering trade-offs and alternatives within competing requirements. Numerical findings obtained through the CRITIC-VIKOR methods indicate that Model 4 stands out as the top-performing alternative with lower Q values and is recognized as the best choice among the options. Conversely, Model 5 is regarded as the least preferred alternative due to its maximum Q value and lowest rank among all alternatives. However, it is crucial to consider the unique characteristics and dynamics of each sport when employing these models. This study's findings show that incorporating fuzzy logic with CRITIC and VIKOR approaches enhances prediction accuracy and reliability for sports events compared to models such as neural networks. The approach effectively handles data complexity, redundancy, and uncertainty, making it a valuable tool for improving decision-making in sports analytics. This study proves that advanced multi-criteria decision-making techniques can lead to more informed and accurate predictions, offering practical benefits for sports analysts and decision-makers. Future research should aim to expand the applicability to different sports and events, incorporate additional variables, and enhance model precision. Such research will contribute to the development of prediction tools that can assist sports professionals, bookmakers, and enthusiasts in making informed decisions.

## Author Contributions

**Conceptualization:** Taibo Liu.

**Writing – original draft:** Taibo Liu.

**Writing – review & editing:** Taibo Liu.

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
