## [Decision Letter · Decision Letter 0]

27 Aug 2024

PONE-D-24-27391Assessing the Effectiveness of Fuzzy Logic-Based Models for Predicting Sports Event Outcomes: A CRITIC-VIKOR ApproachPLOS ONE

Dear Dr. Liu,

Thank you for submitting your manuscript to PLOS ONE. After careful consideration, we feel that it has merit but does not fully meet PLOS ONE’s publication criteria as it currently stands. Therefore, we invite you to submit a revised version of the manuscript that addresses the points raised during the review process.

- What is the difference between use case and applications in the list of contributions?- The authors should clearly highlight the motivation as well as research gap.

We look forward to receiving your revised manuscript.

Kind regards,

Ali Ala

Academic Editor

PLOS ONE

Journal Requirements:

Reviewers' comments:

Reviewer's Responses to Questions

**Comments to the Author**

1. Is the manuscript technically sound, and do the data support the conclusions?

Reviewer #1: Yes

2. Has the statistical analysis been performed appropriately and rigorously? 

Reviewer #1: Yes

3. Have the authors made all data underlying the findings in their manuscript fully available?

Reviewer #1: Yes

4. Is the manuscript presented in an intelligible fashion and written in standard English?

Reviewer #1: Yes

5. Review Comments to the Author

Reviewer #1: (1) The abstract should be improved. A standard abstract must present, without leaving any doubt, the objective of the paper precisely; source of data (which is not present in your abstract) and analytical approach used; key findings and any policy implication and recommendations.

(2) You should provide more recent references published in last two-three years in the Literature review. Remove references published before 2018.

(3) Show step by step algorithm for proposed methodology. You should explain in detail this methodology.

(4) The Introduction chapter is very dense and it contains valuable information. However, the following information should be added to this chapter:

- the research gap: please clearly define and describe the research gap covered by your research proposal;

- the research questions: please try to group them in a distinct paragraph, so that the readers have a clear image on them;

- the research goal.

(5) In the conclusions, the authors simply emphasize the contribution of this paper, while I believe that value judgments based on analysis should be added.

6. PLOS authors have the option to publish the peer review history of their article (what does this mean?). If published, this will include your full peer review and any attached files.

Reviewer #1: No

---

## [Author Response · Author response to Decision Letter 0]

9 Oct 2024

Academic Editor, Concern # 1 (What is the difference between use case and applications in the list of contributions?): 

Author answer: Thank you for the comment: In our study, the use case focuses on a detailed implementation of the CRITIC-VIKOR approach to rank and select the best predictive model among multiple alternatives based on defined criteria such as team performance, historical data, and statistical modeling. 

The paper discusses applications in sports analytics where fuzzy logic-based models can assist coaches, analysts, and team managers in making data-driven decisions by accurately forecasting outcomes. It also mentions potential uses in sports betting, where these models can improve the reliability of predictions, thus impacting decision-making processes within the sports industry. 

Academic Editor, Concern # 2 (The authors should clearly highlight the motivation as well as research gap): 

Author answer: Thank you for valuable comment. Motivation: To improve the precision and reliability of sports outcome predictions by combining fuzzy logic, CRITIC, and VIKOR methods, which better handle data uncertainty and complexity.

 Research Gap: Addressing the lack of adequate, integrated approaches in sports analytics that combine fuzzy logic with multi-criteria decision-making techniques to handle complex, multi-faceted data..

Reviewer#1, Concern # 1 (The abstract should be improved. A standard abstract must present, without leaving any doubt, the objective of the paper precisely; source of data (which is not present in your abstract) and analytical approach used; key findings and any policy implication and recommendations.):

Author answer: Thank you for the valuable comment. The abstract is further improved. 

Author action: We updated the manuscript by ….

Incorporating fuzzy logic-based models into sports prediction has generated significant interest due to the intricate nature of athletic events and the many factors influencing their outcomes. This study evaluates the effectiveness of fuzzy logic-based models in predicting sports event outcomes using a hybrid CRITIC-VIKOR approach. The objective is to improve the accuracy and reliability of sports predictions by addressing the complexity and uncertainty inherent in sports data. The study utilizes a comprehensive dataset comprising historical data on team performance, player statistics, and other relevant factors influencing sports outcomes. The CRITIC method determines each criterion's importance, while the VIKOR method ranks the predictive models to identify the optimal choice. Key findings indicate that the proposed hybrid approach significantly enhances the precision of predictions compared to traditional methods. The best-performing model identified through this approach provides reliable decision support for sports analysts, coaches, and managers. The study recommends incorporating this integrated model into sports analytics for better team management and sports betting decision-making. 

Reviewer#1, Concern # 2 (You should provide more recent references published in last two-three years in the Literature review. Remove references published before 2018):

Author answer: Thank you for the comment. The recent references are included in the reference section. The reference section is further strengthened with the latest publication and the following text added in the literature review to improve the paper.

Author action: We updated the manuscript by ….

Predicting sports outcomes has always been a complex task due to the dynamic nature of sports, the varying performance of players, and the influence of numerous factors such as team strategies, player statistics, and historical data. Accurate predictions can significantly impact decision-making processes in sports management, betting, and team planning [29]. Recent advancements in artificial intelligence (AI) and machine learning have introduced new approaches for enhancing predictive accuracy in sports [30]. Among these, fuzzy logic-based models have gained attention for their ability to handle uncertainty and provide more adaptable predictions in complex environments [31].Combining these models with advanced multi-criteria decision-making techniques like CRITIC (Criteria Importance Through Intercriteria Correlation) and VIKOR (VIšeKriterijumska Optimizacija I Kompromisno Resenje) further enhances their predictive power [32]. 

References

Dell’Anna D, Jamshidnejad A. Evolving fuzzy logic systems for creative personalized socially assistive robots. Eng Appl Artif Intell. 2022;114: 105064.

Rahman M, Tanvir S, Anwar MT. Unique approach to detect bowling grips using fuzzy logic contrast enhancement. In: 2021 IEEE International Conference on Artificial Intelligence in Engineering and Technology (IICAIET); 2021. pp. 1-6.

Hassanniakalager A, Sermpinis G, Stasinakis C, Verousis T. A conditional fuzzy inference approach in forecasting. Eur J Oper Res. 2020;283: 196-216.

Krutikov AK, Meltsov VY, Strabykin DA. Evaluation of the efficiency of forecasting sports events using a cascade of artificial neural networks based on FPGA. In: 2022 Conference of Russian Young Researchers in Electrical and Electronic Engineering (ElConRus); 2022. pp. 355-360.

Author action: We updated the manuscript by ….

Reviewer#1, Concern # 3 (Show step by step algorithm for proposed methodology. You should explain in detail this methodology.):

Author answer: Thank You for the comment. Below is a detailed step-by-step description of the proposed methodology:

Author action: We updated the manuscript by ….

Step 1: Collect a comprehensive dataset that includes historical data on sports events, team and player performance metrics, and other influencing factors.

Step 2: Preprocess the data to ensure quality, consistency, and accuracy, including steps like handling missing values, normalizing data, and preparing the evaluation matrix for analysis.

Step 3: Determine the criteria that will be used to evaluate the predictive models, such as interpretability, historical data reliability, statistical modeling accuracy, and other relevant metrics.

Step 4: Apply the CRITIC Method for Criteria Weighting

4.1 Construct the Evaluation Matrix: Develop a matrix where rows represent the alternatives (predictive models) and columns represent the criteria.

4.2 Normalize the Matrix: Standardize the data to make the criteria comparable, scaling the values between 0 and 1.

4.3 Calculate Correlation Coefficients: Determine the correlation between each pair of criteria to understand their interrelationships.

4.4 Measure the Conflict Between Criteria: Use the correlation data to calculate the measure of conflict for each criterion, which reflects how criteria relate to each other.

4.5 Calculate Criteria Weights: Determine the importance of each criterion using the standard deviation and measure of conflict to assign weights that will be used in the VIKOR method.

Step 5: Rank the Alternatives Using the VIKOR Method

5.1 Construct the Normalized Decision Matrix: Use the normalized values and the weights obtained from the CRITIC method.

5.2 Calculate the Utility (S) and Regret (R) Measures: Compute the utility measure for each model to assess overall performance and the regret measure to evaluate the worst performance of each criterion.

5.3 Calculate the VIKOR Index (Q): Combine the S and R measures to find the VIKOR index for each model, which identifies the compromise solution by balancing overall performance and worst-case scenarios.

5.4 Rank the Models: Rank the alternatives based on their Q values, with the model having the lowest Q value considered the most suitable.

Step 6: Selection of the Optimal Model. Based on the VIKOR ranking, select the model that offers the best balance of utility and regret, effectively addressing the uncertainties and complexities of sports event prediction. Validate the chosen model's performance by comparing it against other alternatives using key metrics.

Reviewer#1, Concern # 4 (The Introduction chapter is very dense and it contains valuable information. However, the following information should be added to this chapter:

- the research gap: please clearly define and describe the research gap covered by your research proposal.):

Author answer: Thank you for the comments. The research gap this study addresses lies in the lack of effective integration of fuzzy logic-based models with advanced multi-criteria decision-making techniques, specifically CRITIC and VIKOR, for predicting sports event outcomes. While traditional methods like machine learning and statistical models have been used for sports predictions, they often fall short in handling the complexity and uncertainty inherent in sports data, such as varying player performance, unpredictable game dynamics, and inconsistent historical data.

Current predictive models frequently rely on single approaches that do not adequately incorporate sports analytics' nuanced, multi-faceted nature. Most of these methods fail to effectively weigh the importance of different criteria or account for the complex interplay between team performance, player statistics, and environmental conditions. A critical gap remains, as a systematic framework is lacking in ranking predictive models and identifying the best alternative..

Our research fills this gap by combining fuzzy logic models with the CRITIC and VIKOR techniques, which together allow for more accurate and reliable predictions. The CRITIC method addresses the need to assess and weight multiple criteria, while the VIKOR method provides a structured approach to ranking models based on a compromise solution. This integration not only improves the predictive accuracy but also offers a comprehensive framework that supports decision-makers in selecting the most suitable model for sports analytics.

Reviewer#1, Concern # 5 (In the conclusions, the authors simply emphasize the contribution of this paper, while I believe that value judgments based on analysis should be added.)

Author answer: Thank you for the comments. The conclusion is now added with value judgment based on analysis, which improved it further and updated the manuscript accordingly.

Author action: We updated the manuscript by

Recently, fuzzy logic-based models have gained attention for predicting sports outcomes, especially in sports forecasting where precise numerical values are difficult to obtain. Accurately predicting results in sports remains a significant challenge. In this research, fuzzy logic-based modeling and the CRITIC-VIKOR method are proposed approaches to addressing the inherent ambiguities and complexities in sports forecasting. The CRITIC method facilitates systematic parameter selection by identifying and prioritizing relevant factors. Meanwhile, the VIKOR method streamlines the selection of appropriate fuzzy logic-based models by considering trade-offs and alternatives within competing requirements. Numerical findings obtained through the CRITIC-VIKOR method indicate that Model 4 stands out as the top-performing alternative with lower Q values and is recognized as the best choice among the options. Conversely, Model 5 is regarded as the least preferred alternative due to its maximum Q value and low rank among all alternatives. However, it is crucial to consider the unique characteristics and dynamics of each sport when employing these models. This study's findings show that incorporating fuzzy logic with CRITIC and VIKOR approaches enhances prediction accuracy and reliability for sports events compared to models such as neural networks. The approach effectively handles data complexity, redundancy, and uncertainty, making it a valuable tool for improving decision-making in sports analytics. This study proves that advanced multi-criteria decision-making techniques can lead to more informed and accurate predictions, offering practical benefits for sports analysts and decision-makers. Future research should aim to expand the applicability to different sports and events, incorporate additional variables, and enhance model precision. Such research will contribute to the development of prediction tools that can assist sports professionals, bookmakers, and enthusiasts in making informed decisions.

---

## [Decision Letter · Decision Letter 1]

3 Nov 2024

Assessing the Effectiveness of Fuzzy Logic-Based Models for Predicting Sports Event Outcomes: A CRITIC-VIKOR Approach

PONE-D-24-27391R1

Dear Dr. Liu,

We’re pleased to inform you that your manuscript has been judged scientifically suitable for publication and will be formally accepted for publication once it meets all outstanding technical requirements.

Kind regards,

Ali Ala

Academic Editor

PLOS ONE

Additional Editor Comments (optional):

Reviewers' comments:

Reviewer's Responses to Questions

**Comments to the Author**

1. If the authors have adequately addressed your comments raised in a previous round of review and you feel that this manuscript is now acceptable for publication, you may indicate that here to bypass the “Comments to the Author” section, enter your conflict of interest statement in the “Confidential to Editor” section, and submit your "Accept" recommendation.

Reviewer #1: All comments have been addressed

2. Is the manuscript technically sound, and do the data support the conclusions?

Reviewer #1: Yes

3. Has the statistical analysis been performed appropriately and rigorously? 

Reviewer #1: Yes

4. Have the authors made all data underlying the findings in their manuscript fully available?

Reviewer #1: Yes

5. Is the manuscript presented in an intelligible fashion and written in standard English?

Reviewer #1: Yes

6. Review Comments to the Author

Reviewer #1: It is ready to be published! All requirements are satisfied. I would like to thank the authors for their nice works

7. PLOS authors have the option to publish the peer review history of their article (what does this mean?). If published, this will include your full peer review and any attached files.

Reviewer #1: **Yes: **Serhat Yüksel

---

## [Editor Report · Acceptance letter]

10 Dec 2024

PONE-D-24-27391R1 

PLOS ONE

Dear Dr. Liu, 

I'm pleased to inform you that your manuscript has been deemed suitable for publication in PLOS ONE. Congratulations! Your manuscript is now being handed over to our production team.

Kind regards, 

on behalf of

Dr. Ali Ala 

Academic Editor

PLOS ONE